# Infrared and Visible Image Fusion with Significant Target Enhancement

**DOI:** 10.3390/e24111633

**Published:** 2022-11-10

**Authors:** Xing Huo, Yinping Deng, Kun Shao

**Affiliations:** 1School of Mathematics, Hefei University of Technology, Hefei 230009, China; 2School of Software, Hefei University of Technology, Hefei 230009, China

**Keywords:** image fusion, infrared image, visible image, significant target enhancement, multi-level Gaussian curvature filtering, ResNet50

## Abstract

Existing fusion rules focus on retaining detailed information in the source image, but as the thermal radiation information in infrared images is mainly characterized by pixel intensity, these fusion rules are likely to result in reduced saliency of the target in the fused image. To address this problem, we propose an infrared and visible image fusion model based on significant target enhancement, aiming to inject thermal targets from infrared images into visible images to enhance target saliency while retaining important details in visible images. First, the source image is decomposed with multi-level Gaussian curvature filtering to obtain background information with high spatial resolution. Second, the large-scale layers are fused using ResNet50 and maximizing weights based on the average operator to improve detail retention. Finally, the base layers are fused by incorporating a new salient target detection method. The subjective and objective experimental results on TNO and MSRS datasets demonstrate that our method achieves better results compared to other traditional and deep learning-based methods.

## 1. Introduction

Image fusion is an image enhancement technique that aims to integrate complementary and redundant information from multiple sensors, which can generate more reliable, robust, and informative images. Infrared and visible image fusion is an important field in image fusion [1]. Infrared images capture the thermal radiation emitted in the scene, which can effectively distinguish the target and background in the scene, and is not affected by occlusions and extreme weather. However, infrared images have shortcomings such as not obvious details [2]. According to this characteristic of infrared images, many different areas have been researched to identify prominent targets in infrared scenes, such as infrared image segmentation [3], pedestrian detection [4], etc. Visible images focus on capturing reflected light, which has quite high spatial resolution and contrast. Although visible images can effectively preserve the details of the scene, they are vulnerable to factors such as light intensity and extreme weather. The infrared and visible image can not only highlight the thermal radiation information in the infrared image, but also get the rich details in the visible image. Therefore, it is widely used in many fields such as military detection and public security [5].

In general, existing image fusion frameworks mainly include multi-scale transform [6], sparse representation [7], deep learning [8], etc. Currently, the most widely employed method for infrared and visible image fusion is the multi-scale transform, which can decompose the source image into multiple sub-images at different scales and effectively retain the detail features at different scales, so that the fused image can obtain a good visual effect. Traditional image fusion based on multi-scale transform includes pyramid transform [9], discrete wavelet transform [10], curvelet transform [11], and so on. These multi-scale transform methods can fuse the source images at different scales, but there are still some defects. For example, the pyramid transform does not have translation invariance, which may lead to a false Gibbs phenomenon in the fused image. Wavelet and curvelet transform have been improved, but they do not fully take into account the spatial consistency, which may lead to the color brightness distortion of the fusion results [12]. To improve these problems, the edge preserving filters are proposed, such as guide filter [13] and bilateral filter [14], which are very effective in solving the spatial consistency problem and can reduce artifacts around edges [15]. In recent years, some new multi-scale methods have achieved good results. More specifically, Jia et al. [16] proposed a stretched multi-scale fusion method for infrared and visible images, which can improve the information content of fusion results. Li et al. [17] proposed a multi-exposure fusion method for alleviating the issue of lost details in fusion results that may be caused by traditional multi-scale. Meanwhile, hybrid filters have also been widely used in the field of image fusion. Do et al. [18] proposed a method combining Gaussian filtering (GF) and rolling guidance filter (RGF), which solved the problem that conventional multi-scale methods for image decomposition could only be performed from a single channel. Zhou et al. [9] proposed a hybrid multi-scale decomposition method combining GF and bilateral filtering, which improved the problem of unstable bilateral filtering weights. The Gaussian curvature filtering (GCF), an edge-preserving filter proposed in recent years, is superior to RGF and other edge-preserving filters because of its excellent parameter-free characteristics and high efficiency of fine-scale retention. Although the above methods have achieved certain effects, there are still some problems, such as not considering the spatial scale may lead to the loss of some details, and cannot be well extracted salient targets.

In recent years, deep learning methods have developed rapidly in the field of image fusion. Liu et al. [19] used CNN for multi-focus image fusion and achieved good fusion performance in both subjective and objective evaluation. However, this method only deployed the results of the last layer as image features, which may lose a lot of useful information obtained from the middle layer. Li et al. [20] applied VGG to an infrared and visible image fusion task, using VGG19 to reconstruct the detailed content and retain as much detailed information as possible. However, the deep learning fusion method based on the above does not make full use of the deep features, and with the deepening of these network layers, the performance will tend to saturate or even decline. To address these issues, Chen et al. [21] proposed an attention-guided progressive neural texture fusion model to suppress noise in the fusion results. Bai et al. [22] developed an end-to-end deep pre-dehazer model. Moreover, Li et al. [23] devised a meta learning-based deep framework for fusing infrared and visible images with different spatial resolutions. Ma et al. proposed DDcGAN [24], an end-to-end model for fusing infrared and visible images of different resolutions. Li et al. proposed DenseFuse [25] and RFN-Nest [26], infrared and visible image fusion frameworks based on self-Auto Encoder. Xu et al. proposed an unsupervised end-to-end image fusion network, U2Fusion [27], which achieved good results. However, these methods still have some defects in target saliency extraction, such as they cannot well highlight infrared salient targets.

To solve the above problems, a new infrared and visible image fusion framework is proposed, which preserves the details as much as possible on the basis of considering the spatial scale, reduces the generation of artifacts, and improves the saliency of the target. Since multi-level Gaussian filtering (MLGCF) [28] can better obtain high spatial resolution background information, we use it to decompose the source image into large-scale layer, small-scale layer, and base layer. For large-scale layers, ResNet50 is used to extract features, then ZCA and L1 norms are used to construct the fusion weights. The max-absolute rule is adopted for the small-scale layer. A new frequency tuning saliency detection method (FT++) is proposed for base layer fusion. The experimental results on TNO and MSRS datasets show that our method outperforms many state-of-the-art methods.

The main contributions of this paper are as follows:(1)We propose a novel infrared and visible fusion method, which can effectively retain detailed information and maintain the target saliency. This method can be widely used in the military, target detection, and other fields.(2)More abundant details can be obtained from the source image by employing MLGCF and ResNet50 to extract features.(3)A new approach to constructing saliency map (FT++) is proposed, which can productively retain the thermal radiation information. Extensive qualitative and quantitative experiments demonstrate the superiority of our method compared to the latest alternatives. Compared with other competitors, our approach could generate fused images looking like high-quality visible images with highlighted targets.

The remainder of this paper is organized as follows. Section 2 presents the theory related to residual networks and FT saliency detection. Section 3 describes the proposed strategy for infrared and visible image fusion in detail. In Section 4, our approach is compared with some existing state-of-the-art methods on two different datasets, and ablation experiments as well as fusion performance analysis are performed to verify the effectiveness of the proposed method. Finally, conclusions are presented in Section 5.

## 2. Correlation Theory

### 2.1. Residual Network

He et al. [29] proposed a residual network and introduced jumping connection lines on each residual block structure, which solved the problems of gradient disappearance caused by the increase of network layers and rapid decline after accuracy saturation. Compared with VGG19, the residual network has a superior ability to extract features. Considering the complexity of the task and the size of the data, we leverage ResNet50, a network of 50 weight layers, for the extraction of detailed features. The structure of the residual blocks in ResNet50 is shown in Figure 1:

Where X represents the input of residual block structure, F(X) represents the result of input X calculated by two weight layers, and ψ(X)=F(X)+X is the feature learned by the residual block structure. The core idea of the network is to introduce a jump connection line to ensure that the output ψ(X) can at least learn new features. When the gradient disappears, ψ(X)=X is an identity map, and many such structures are stacked together, ensuring that the result of this network is at least as good as that of the shallow network.

### 2.2. FT Significance Detection

Achanta, R. et al. [30] proposed a saliency detection algorithm based on FT. The main principle of the algorithm is to discard the high-frequency information in the frequency domain and retain the low-frequency information such as the contour and basic composition of the image as much as possible. The mathematical expression of pixel saliency is as follows:(1)$$S(p)=\left\|I_{\mu }-I_{whc}(p)\right\|,$$
where $I_{\mu }$ is the average pixel value of input image I. $I_{whc}(p)$ is the pixel value of input image I at point p, that processed by GF with a window size of 5×5, and $\left\|\cdot \right\|$ represents the L2-norm.

FT algorithm carries out saliency detection from the perspective of spatial frequency, which has simple and efficient advantages. However, because the original FT algorithm employs GF to process the input image, the strong and weak edge information of the image is easy to be blurred by GF, and the key information of the image cannot be fully extracted. To solve these problems, in recent years, Hou et al. improved the FT algorithm by guide filter [31]. As an edge-preserving filter, guide filter uses the mean and variance of pixel neighborhood as local estimation, which has a strong edge-preserving ability. However, considering that RGF as a hybrid filter, it combines the smoothing characteristics of GF and the edge retention characteristics of guide filter. Therefore, it can extract information from different scales and has a stronger edge preserving ability compared with GF and guide filter [15]. Based on the above analysis, we adopt RGF to replace GF to improve original FT (FT++).

The FT++ method is shown in Figure 2. First, the source image is processed with RGF and converted from RGB to LAB color space to obtain $I_{rgf}$, which can be represented as $I_{rgf}(x,y)={\left[l_{rgf},a_{rgf},b_{rgf}\right]}^{T}$, where $l_{rgf}$, $a_{rgf}$, $b_{rgf}$ represent the three channels in LAB color space, respectively. Second, $I_{\mu }={\left[l_{\mu },a_{\mu },b_{\mu }\right]}^{T}$ is obtained by calculating the average value for each channel. Finally, the pixel significant values are acquired according to Equation (7).

We compare FT++ with the original FT method to verify the effectiveness of the method, see Figure 3. It can be seen from Figure 3d that there are many edge features such as people and houses in the difference map [32]. Therefore, FT++ can obtain more salient information than the original FT method.

## 3. Proposed Fusion Framework

The infrared and visible image fusion based on multi-scale decomposition and FT saliency can effectively retain the texture details of the source image and enhance significant targets. After decomposing the source image using MLGCF, the base layer saliency map is extracted by FT++, the large-scale features are processed by introducing ResNet50 etc., and finally the final fused image is reconstructed. The flow chart of the method in this paper is shown in Figure 4.

### 3.1. Image Decomposition

The edge-preserving filter can improve the common problems in the multi-scale decomposition process, such as the tendency to produce halo artifacts [33], and does not take full account of space consistency, which can lead to distorted color brightness of the fusion results. To solve these problems, we adopt MLGCF to decompose the source image. GF is a classical tool for image degradation and smoothing. GCF is an effective edge-preserving filter with the advantage of being parameter-free and retaining fine detail. It is instructive to note that MLGCF takes use of the smoothing properties of the GF and the edge-preserving properties of the GCF to obtain features at different scales. The specific process is divided into the following three steps:

(1)Using GF to smooth small structure information:
(2)$$I_{k,g}=Gaussian(I_{k},\sigma_{s}),$$
where $I_{k}(k\in 1,2)$ is the input image. $I_{1}$ and $I_{2}$ are infrared image and visible image respectively. $I_{k,g}$ is the result of the input image processed by GF. $\sigma_{s}$ is the standard deviation of GF, which is mainly used to smooth the texture details of the image.(2)Using GCF for the edge recovery process:
(3)$$I_{k,gcf}=GCF(I_{k},m),$$
the parameter m is the number of iterations, and we set m=5 based on experience.(3)Combining GF with GCF using a hybrid multiscale approach for a three-stage decomposition:

(4)$$D_{k}^{i,1}=\left\{\begin{array}{l} I_{k}-I_{k,gcf}^{1},\;\;\;\;i=1\;\\ I_{k,g}^{i-1}-I_{k,gcf}^{i},\;\;i=2,3\; \end{array}\right.,$$(5)$$D_{k}^{i,2}=I_{k,gcf}^{i}-I_{k,g}^{i},\;\;\;\;i=1,2,3,$$(6)$$B_{k}=I_{k,g}^{3},\;\;\;\;i=3,$$
where $D_{k}^{i,j}(j=1,\;2)$ represents the texture detail and edge detail on the multi-scale decomposition of layer i, respectively, i∈{1,2,3} denotes the number of decomposition layers. Record the result of the last GF decomposition as base layer $B_{k}$.$I_{k,g}^{i}(i=1,2,3)$ represents the result of the *i*-th GF process of the input image $I_{k}$.$I_{k,gcf}^{i}(i=1,2,3)$ denotes the result of $I_{k}$ after the *i*-th GCF process. The parameter $\sigma_{s}$ is the variance in the GF operation and is taken as $\sigma_{s}=20$ in this paper.

To verify the advantages of the MLGCF decomposition, the MLGCF is compared with the RGF decomposition. As can be seen in Figure 5, the RGF decomposition has a halo around the target as the number of layers deepens, but this is barely visible with the MLGCF decomposition. Furthermore, the MLGCF decomposition results show that each scale layer contains the specific content of the current detail layer. Therefore, the MLGCF decomposition has the effect of suppressing haloing and preserving the content of a specific scale of detail.

### 3.2. Image Fusion

#### 3.2.1. Fusion Strategy for the Base Layer

In the past, most of the base layers were fused using simple averaging or weighted averaging, although these methods are simple to operate, they tend to lead to problems such as poor target saliency and low overall contrast of fusion results. To solve these problems, we adopt the FT++ method to process the infrared base layer and deploy its normalized result as the fusion weight. The specific steps are as follows:

(1)FT++ method: The FT++ method in this paper only processes infrared images, so the input image for this process is the infrared image $I_{1}$. An improvement is made using the RGF instead of the GF in the original FT algorithm, as shown in Figure 2.

Calculating saliency map:(7)$$S(p)=\left\|I_{\mu }-I_{rgf}(p)\right\|,$$
$I_{\mu }$ is the average pixel value of infrared image $I_{1}$. $I_{rgf}(p)$ is the pixel value of $I_{1}$ at point p after RGF processing.

(2)Normalizing the significance map to obtain the base layer fusion weights $W_{b}$:



(8)
$$W_{b}=\frac{S(p)}{\max(S(p))}.$$



(3)Fusion of base layers using a weighted average strategy:
(9)$$F_{b}=W_{b}\cdot B_{1}+(1-W_{b})\cdot B_{2},$$
where $B_{k}(k=1,2)$ is the base layer of the infrared and visible images respectively and $F_{b}$ is the fusion result of the base layer.

#### 3.2.2. Fusion Strategy for Small-Scale Layers

The texture information of the source image is contained in the small-scale layer. Usually, the larger the pixel value of the small-scale layer, the more texture information is retained [28], so for the fusion of the small-scale layer, we leverage the “Max-absolute” fusion method. The small-scale texture details and edge details are $D_{k}^{1,1}$, $D_{k}^{1,2}$, respectively.

Small-scale texture detail fusion results:(10)$$F_{1}^{1}(x,y)=\max(D_{1}^{1,1}(x,y),D_{2}^{1,1}(x,y)).$$

Small scale edge detail fusion results:(11)$$F_{1}^{2}(x,y)=\max(D_{1}^{1,2}(x,y),D_{2}^{1,2}(x,y)).$$

#### 3.2.3. Fusion Strategy for Large-Scale Layers

The edge information and structural information of the source image are contained in the large-scale layer. Although deep learning fusion methods can effectively extract deep features, most of them only extract features without processing the extracted features, which may lead to the degradation of fusion results [34]. In order to make full use of and process the useful information in the deep network, and considering the complexity and data scale of the task, ResNet50 with ImageNet fixed training is used in this paper to extract large-scale layer features [29]. Then ZCA and L1-norm are employed to normalize the extracted features. Finally, the fusion weight is constructed to obtain the large-scale layer fusion results. The overall process is as follows:

(1)Feature extraction: First, the large-scale layer $D_{k}^{i,j}(i=2,3)$ is input into ResNet50 to extract features. The texture features and edge features extracted to layer i(i=2,3) are denoted as $F_{k}^{i,j,t,c}(j=1,2)$, where t(t=1,2,⋯,5) denotes the *t*-th convolutional block, and we take t=5. c(c=1,2,⋯,C) denotes the *c*-th channel of the output feature, and C is the number of channels at level t, $C=64\times 2^{t-1}$.(2)The extracted features are ZCA processed to obtain the new features ${\hat{F}}_{k}^{i,j,t,c}$, then the L1-norm of ${\hat{F}}_{k}^{i,j,t,c}$ is calculated, and finally, we deploy the average operator to calculate the activity level measurement:
(12)$$C_{k}^{i,j,t}(x,y)=\frac{\sum_{\beta =-r}^{r}\sum_{\theta =-r}^{r}{\left\|{\hat{F}}_{k}^{i,j,t,1:C}(x+\beta ,y+\theta )\right\|}_{1}}{{\left(2r+1\right)}^{2}},$$
where the size of r determines the size of the extracted image block in the new feature ${\left\|{\hat{F}}_{k}^{i,j,t,1:C}\right\|}_{1}$. When r is too large, detail information may be lost [25], so we take r=1.(3)Construction of initial weight maps using Softmax:
(13)$${\hat{C}}_{k}^{i,j,t}(x,y)=\frac{C_{k}^{i,j,t}(x,y)}{\sum_{k=1}^{2}C_{k}^{i,j,t}(x,y)}.$$(4)Using a maximum weight construction method based on average operator (MWAO) method: In order to obtain as much detail information as possible, the largest pixel value in Equation (13) is taken on each large-scale layer as the fusion weight for that layer. Finally, the obtained weight is used to reconstruct the large-scale layer of fusion image:



(14)
$$W_{k}^{i,j,t}=\max({\hat{C}}_{k}^{i,j,t}(x,y)).$$


(15)
$$F_{i}^{j}=W_{1}^{i,j,t}\cdot D_{1}^{i,j}+W_{2}^{i,j,t}\cdot D_{2}^{i,j}(i=2,3\;,\;j=1,2).$$



The MWAO method is compared with the method of constructing weight map [34] to verify its superiority. As shown in Table 1, it can be seen that the method of selecting the maximum weight to construct the fusion weight has more advantages than the original scheme of using the weight map in objective evaluation.

### 3.3. Reconstructing Fusion Image

Reconstruction of the fused image using the obtained fused base layer $F_{b}$ and the detail layer $F_{i}^{j}(i=1,2,3\;,\;j=1,2)$:(16)$$F=F_{b}+\sum_{j=1}^{2}\sum_{i=1}^{3}F_{i}^{j}.$$

## 4. Experimental Results and Comparisons

This section first introduces the datasets and evaluation metrics, as shown in Section 4.1 and Section 4.2, respectively. Then we make a quantitative and qualitative comparison with the state-of-the-art methods, as shown in Section 4.3 and Section 4.4 respectively. Finally, in Section 4.5 and Section 4.6, the rationality and superiority of the method were proved by the ablation experiment and fusion performance analysis. We will introduce each part in detail in the following. The CPU used for the experiment is Intel Core i7-11800H, the graphics card is NVIDIA RTX 3060, the operating system is Windows 10, and the programming software is Matlab2016b.

### 4.1. Experimental Datasets

Subjective and objective evaluations of our method were carried out on two different datasets. The datasets are derived from TNO [35] and MSRS [36], and the selected images are aligned. Among them, the TNO dataset contains infrared and visible images of different military scenes, and MSRS dataset contains multiple infrared and visible images of multi-spectral road scenarios. In the subjective evaluation, five groups of representative infrared and visible images were selected for comparison on the two datasets, among which Un Camp, Kaptein_1123, Bench, Tree, and Pavilion were selected for TNO and 00352D, 00196D, 00417D, 00427D, 00545D were selected for MSRS, as shown in Figure 6 and Figure 7. In the objective evaluation, 20 groups of registered infrared and visible images were selected from each of the two datasets to calculate the corresponding evaluation metrics values. The detailed experimental results are shown in the Figure 8, Figure 9, Figure 10 and Figure 11.

### 4.2. Fusion Metrics

In order to reduce the interference of human consciousness in subjective evaluation, we chose six evaluation metrics, namely Entropy (EN) [37], Standard Deviation (SD) [38], Average Gradient (AG) [39], Visual Information Fidelity (VIF) [40], Mutual Information (MI) [41], and Cross Entropy (CE) [42], to validate the validity and superiority of our proposed method.

EN computes the amount of information contained in the fused image based on information theory. The higher the EN, the richer the information contained in the fused image:(17)$$EN=-\sum_{i=0}^{L-1}p_{i}\log_{2}p_{i},$$
where L is the number of grey levels, $p_{i}$ is the normalized histogram of the corresponding gray level in the fused image.

SD is used to describe the statistical distribution and contrast features of images. The larger the SD, the higher the image contrast:(18)$$\mathrm{SD}=\sqrt{\sum_{i=1}^{M}\sum_{j=1}^{N}{\left(F(i,j)-\mu \right)}^{2}},$$
where F is the image fusion result, and μ is the average pixel value of the fusion result.

AG is used to reflect the sharpness of the image. The larger the AG, the clearer the texture of the details contained in the image:(19)$$\begin{array}{c} AG=\frac{1}{M\cdot N}\sum_{i=1}^{M}\sum_{j=1}^{N}\sqrt{\frac{\nabla F_{x}^{2}(i,j)+\nabla F_{y}^{2}(i,j)}{2}},\\ \nabla F_{x}(i,j)=F(i,j)-F(i+1,j),\\ \nabla F_{y}(i,j)=F(i,j)-F(i,j+1). \end{array}$$

VIF can objectively express people’s feelings when observing images. The larger the VIF, the better the visual effect of the images. The building process is divided into four steps: First, the two source images and their fusion results are divided into blocks; second, evaluating the visual information of the first step’s block results, both with and without distortion; third, calculating the VIF of each sub-band; fourth, calculate the overall indicators.

MI indicates the amount of information obtained from the source image, the larger the MI, the more information is obtained.
(20)$${\mathrm{MI}}_{X,F}=\sum_{x,f}p_{X,F}(x,f)\log\frac{p_{X,F}(x,f)}{p_{X}(x)p_{F}(f)},$$
(21)$$MI=MI_{I,F}+MI_{V,F},$$
where $p_{X,F}$ represents the joint probability density, $p_{X}$, $p_{F}$ represent the edge probability density, $MI_{I,F}$ and $MI_{V,F}$ represent the information content of infrared image and visible image respectively.

CE is the average value of the relative entropy between the two source images and the fused image, which is used to characterize the pixel difference at the corresponding position between the two source images and the fused image [42]. The smaller the CE, the better the image fusion effect:(22)$$\begin{array}{c} D(p∥q)=\sum_{i=0}^{L-1}p(i)\log_{2}\frac{p(i)}{q(i)},\\ CE(I,V,F)=\frac{D(h_{I}∥h_{F})+D(h_{V}∥h_{F})}{2}, \end{array}$$
where p(i) and q(i) are two probability distribution functions, $h_{I}$, $h_{V}$, and $h_{F}$ are the normalized histograms of infrared image, visible image and source image, respectively.

### 4.3. Subjective Evaluation

Figure 8 and Figure 9 show the comparison results of our method with the nine methods on TNO and MSRS datasets, respectively. The nine methods are as follows: (a) GTF [43], (b) WLS [44], (c) MLGCF [28], (d) ResNet50 [34], (e) RFN -Nest [26], (f) DenseFuse [25], (g) U2Fusion [27], (h)FusionGAN [45], (i)GANMcC [46]. Where (a–c) are representative of traditional methods, (d–i) are advanced deep learning methods in recent years, and (j) is our method. Targets that are not significant in the comparison methods are marked with red rectangles, and poor details are marked with green rectangles.

#### 4.3.1. Subjective Evaluation on the TNO Datasets

The subjective evaluation of fusion results on the TNO dataset is shown in Figure 8. As can be seen, in Figure 8I, the low contrast of the person in (a,d,e–g) indicates that some infrared information was lost during the fusion process. The color of the leaves in (b,e–h) are inconsistent and there are obvious abrupt changes. The eaves and fences of (a,c,i) are not clear, indicating that some details are missing. The overall visual fidelity in (a,h,i) is low, resulting in distortion of the person. By contrast, our method (j) not only can better highlight the target person, but also retain detail information such as leaves and fences.

In Figure 8II–V, the (d–g) have less significant target, and the (a) is more toward the infrared images. Specifically, in Figure 8IV, the bright grass in (a) is not extracted in the place circled by the red box, the dark trees in (b–d,h,i) are not highlighted, which makes it difficult to see the specific background information as a whole. In Figure 8V, the leaves of bushes in (a,d,e,h,i) are not clear and do not highlight the gradually bright characteristics of bushes from the lower left to the upper right. Compared with other methods, our method performs well, especially in detail extraction and target saliency. However, the infrared targets in (b,c,f) of Figure 8II,IV are more natural compared to (j), so further validation of the fusion effect using objective quality evaluation is needed.

#### 4.3.2. Subjective Evaluation on the MSRS Datasets

Figure 9 shows our results on MSRS. It can be seen that the target significance of (a) and (d–g) is low. The sky colors of (a) and (d–h) in Figure 9I show abrupt changes, which do not conform to human visual observation. The pipe details of (a–i) and the ground texture of (d) and (f–h) in Figure 9II are not clear. In Figure 7III, there is obvious noise in (g), and the road color in (a,e,f,h,i) is closer to the infrared image. In Figure 9IV, the contrast between (d,e,h,i) is low, and the water beach is not obvious. In Figure 9V, the road surface (b,c) appears noisy, and the vehicles (a,d,e) are blurred. In contrast, our method shows great fusion effects, especially in terms of contrast enhancement and highlighting the target.

### 4.4. Objective Evaluation

#### 4.4.1. Objective Evaluation on the TNO Datasets

We employ the six evaluation metrics mentioned in Section 4.2 to objectively evaluate 20 groups of infrared and visible images on the TNO dataset, and the results are shown in Figure 10. The values in the legend represent the average of the metrics after removing the maximum and minimum values.

Figure 10 shows that our method is superior to other comparison methods, which is consistent with the subjective evaluation results. The significant improvement in EN indicates that our method performs well in terms of information retention. The improvement of SD and VIF indicates that the fusion results of our method have high contrast and good visual effect. This is because we develop FT++ to obtain the saliency map of the infrared base layer. The increase in the AG indicates an improvement in the clarity of the fusion results. The improvement in the MI indicates that the fusion results are rich in information from the source images. The reduction in CE indicates that the difference between our fusion results and the two source images is smaller. All these are due to the combined use of our decomposition method and fusion rules, which gives the overall method a significant advantage in terms of both information retention and target saliency enhancement.

#### 4.4.2. Objective Evaluation on the MSRS Datasets

Figure 11 shows our objective evaluation results on the MSRS dataset. It can be seen that our method is significantly better than the other seven methods in the six evaluation metrics of EN, SD, AG, VIF, MI and CE, which indicates that our method has high contrast and obtains rich information from the source image. Although our method is not optimal in AG, it is still better than most methods. Combined with the results of the subjective evaluation, our method is visually excellent and shows an outstanding competitive advantage.

### 4.5. Ablation Experiments

To demonstrate that the methods can produce beneficial effects, we conduct ablation experiments for the base and large-scale layer methods separately. The experiment consists of six parts: (i) Removing the large-scale layer (ResNet50 and MWAO, etc.,) fusion method from this paper and using FT++ for the base layer; (ii) removing the large-scale layer fusion method and using FT for the base layer; (iii) removing all three methods; (iv) keeping the large-scale layer fusion method from this paper only; (v) keeping the large-scale layer fusion method and using FT for the base layer; (vi) keeping the large-scale layer fusion method and using FT++ for the base layer (our method), where the removed methods are replaced with those of the corresponding scales from the Ref. [28]. Table 2 shows the objective evaluation metric values after the average value of the 20 groups of image fusion results on TNO datasets.

It can be seen that our method achieves the optimum in EN, SD, AG, and CE metrics, which shows that our method has great advantages in information and contrast. The main reason for the lack of advantages in VIF and MI is that our method discarded unnecessary redundant information in the fusion process, which led to the reduction of some reference-based evaluation metrics. However, combined with the subjective evaluation results, our method has clear details and prominent targets, so it provides a good visual effect. In addition, it can be seen from the values of each evaluation metric that the FT and FT++ methods have higher metric values, and among these methods, the combination of ResNet50 and FT++ methods have the best overall performance. This phenomenon shows that the method achieves a better fusion effect overall. It can therefore be shown that the FT++ method and the MWAO method based on the average operator help to improve the image fusion quality.

### 4.6. Fusion Performance Analysis

Based on the above subjective and objective evaluation, it can be seen that our method is significantly better than other methods, which proves that our method can more effectively obtain high-quality fused images.

Because the hybrid multi-scale decomposition and ResNet50 fusion rule are relatively time-consuming, the running time of our method is slightly longer than that of other traditional methods. However, in terms of fusion effect, compared with the optimal method in the comparison method on TNO dataset, the EN, SD, AG, VIF, MI, and CE of our method are improved by 3.82%, 29.78%, 5.47%, 14.96%, 3.82%, and 30.41% on average, respectively. On MSRS dataset, the EN, SD, VIF, MI, and CE of our method are improved by 5.79%, 14.06%, 24.46%, 5.79%, and 17.51% on average, respectively. Analyzing the above time and performance together, our method is far superior in performance to other comparative methods. Therefore, the cost of a reasonable increase in running time is feasible and worthwhile in order to obtain better fusion results for precise and widespread application in various fields.

## 5. Conclusions

In this study, we design a novel infrared and visible image fusion method based on significant target enhancement. The proposed method solves the problem regarding the preservation of thermal radiation features. MLGCF is deployed to decompose the source image and extract useful information accurately. To provide smooth and prominent base layers for fusion results, we propose a new method (FT++) to construct the fusion weights for the base layer. Large-scale features are extracted and processed by ResNet50 and ZCA in such a way as to preserve useful details in the source images effectively. The subjective and objective comparison results on TNO and MSRS datasets demonstrate that our method achieves better results compared to the traditional and deep learning-based alternatives. Although our method has shortcomings in terms of running efficiency, the fusion results are improved significantly over other approaches. In upcoming future research, we will further improve this method by reducing the time consumption and deploy it to the target detection task.

## Figures and Tables

**Figure 1 entropy-24-01633-f001:**
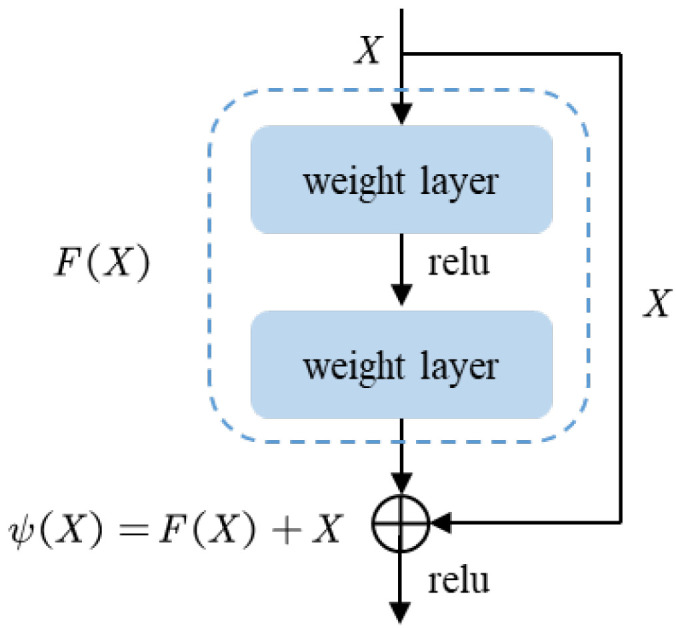
Residual block structure.

**Figure 2 entropy-24-01633-f002:**
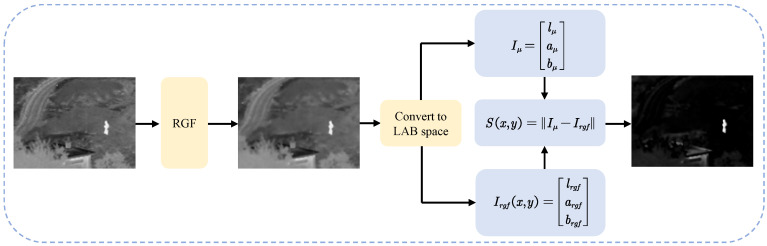
Flowchart for extracting salient targets with FT++.

**Figure 3 entropy-24-01633-f003:**
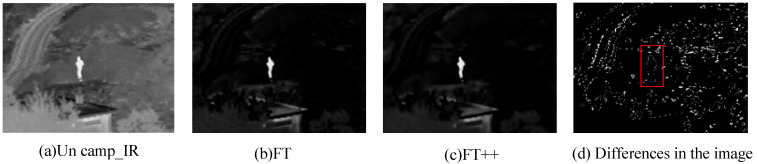
Comparison of different feature extraction methods for infrared images.

**Figure 4 entropy-24-01633-f004:**
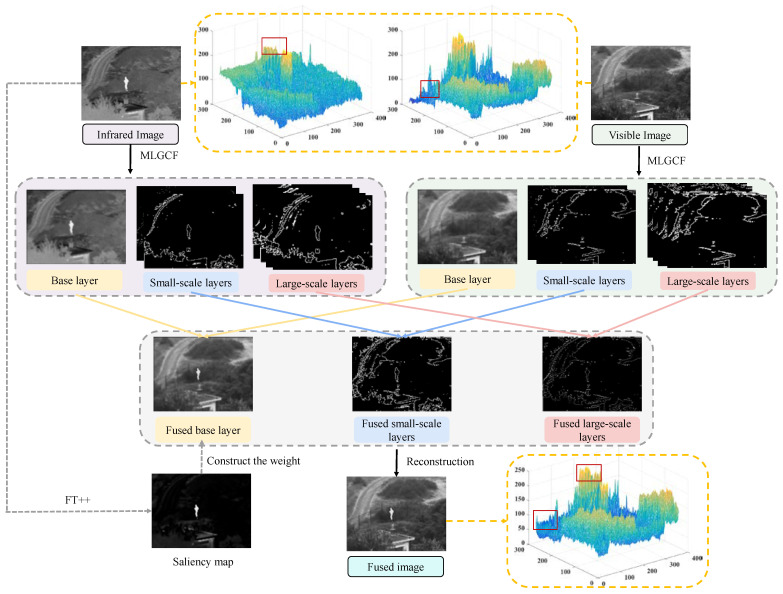
The image fusion framework of our method.

**Figure 5 entropy-24-01633-f005:**
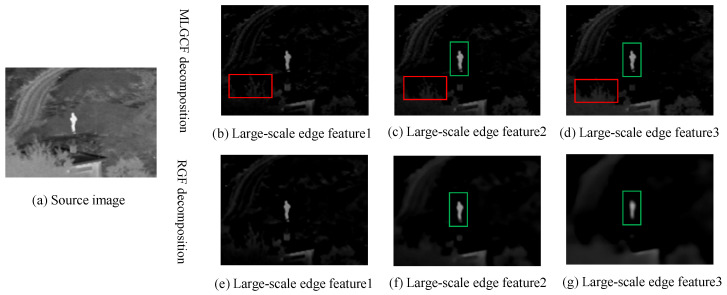
Detail comparison of MLGCF and RGF.

**Figure 6 entropy-24-01633-f006:**
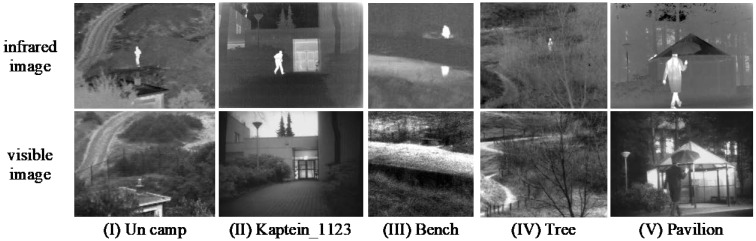
Five typical infrared and visible images on the TNO dataset.

**Figure 7 entropy-24-01633-f007:**
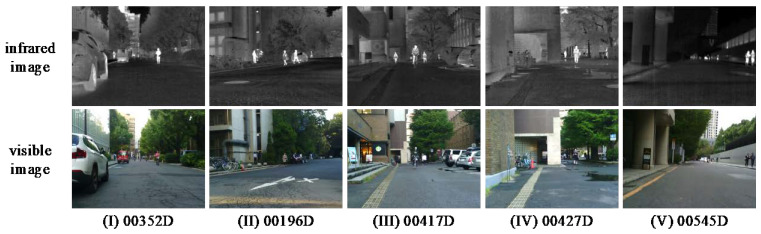
Five typical infrared and visible images on the MSRS dataset.

**Figure 8 entropy-24-01633-f008:**
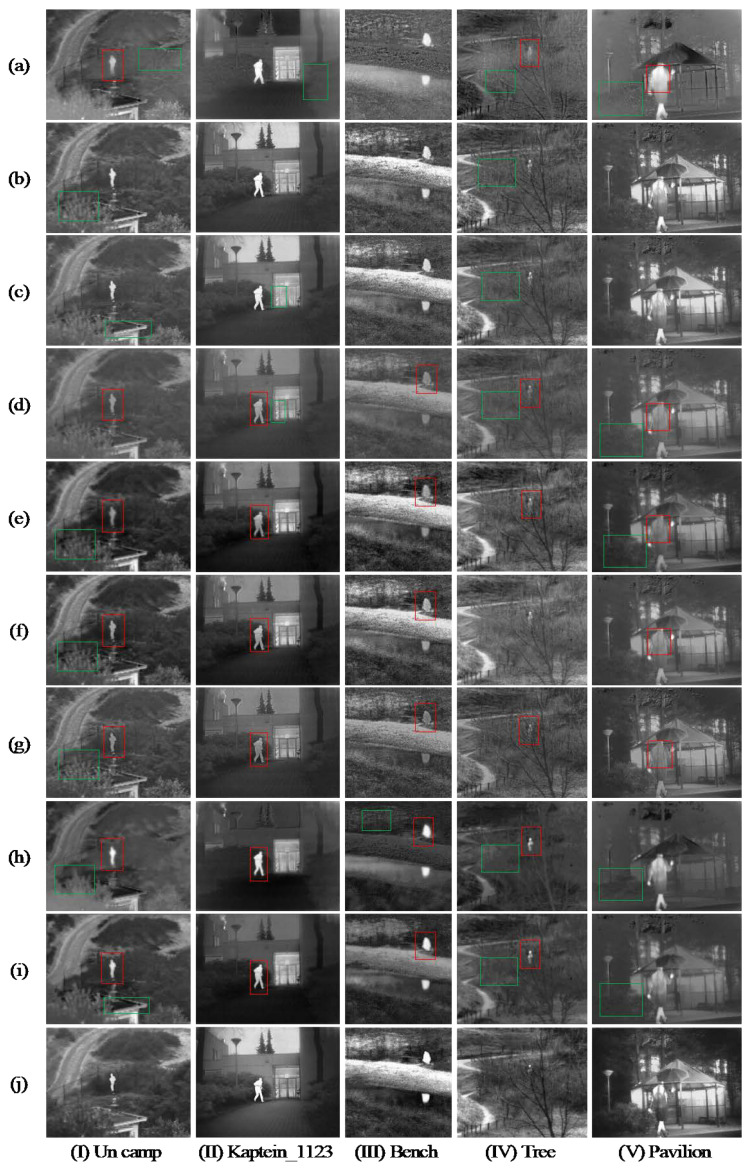
Typical fusion results of five infrared and visible images on the TNO dataset. (**a**) GTF, (**b**) WLS, (**c**) MLGCF, (**d**) ResNet50, (**e**) RFN-Nest, (**f**) DenseFuse, (**g**) U2Fusion, (**h**) FusionGAN, (**i**) GANMcC, (**j**) Ours.

**Figure 9 entropy-24-01633-f009:**
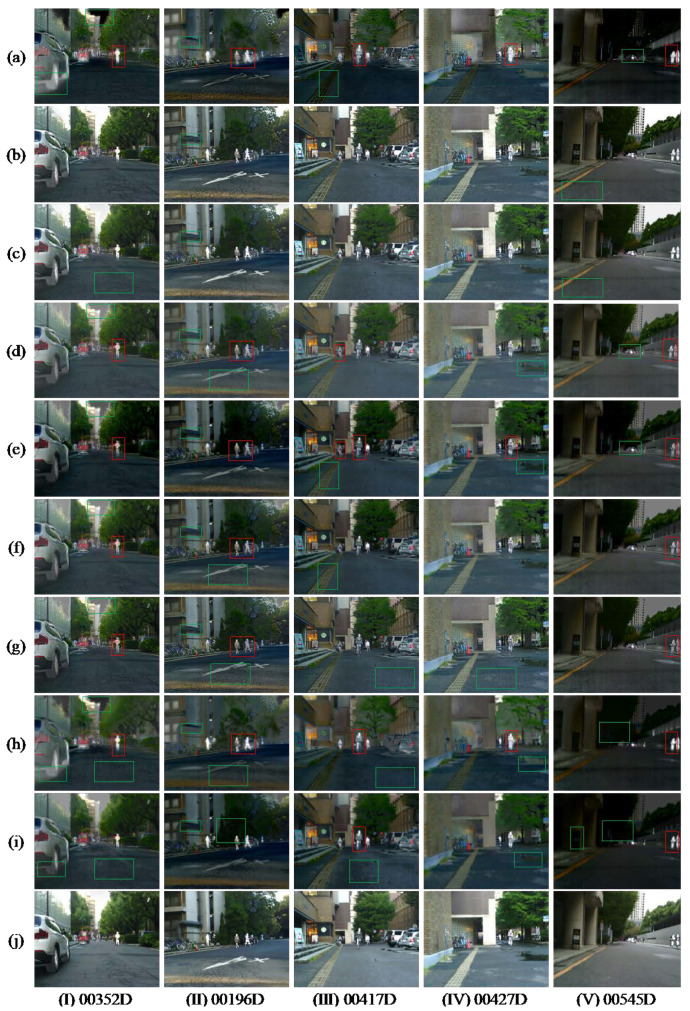
Typical fusion results of five infrared and visible images on the MSRS dataset. (**a**) GTF, (**b**) WLS, (**c**) MLGCF, (**d**) ResNet50, (**e**) RFN-Nest, (**f**) DenseFuse, (**g**) U2Fusion, (**h**) FusionGAN, (**i**) GANMcC, (**j**) Ours.

**Figure 10 entropy-24-01633-f010:**
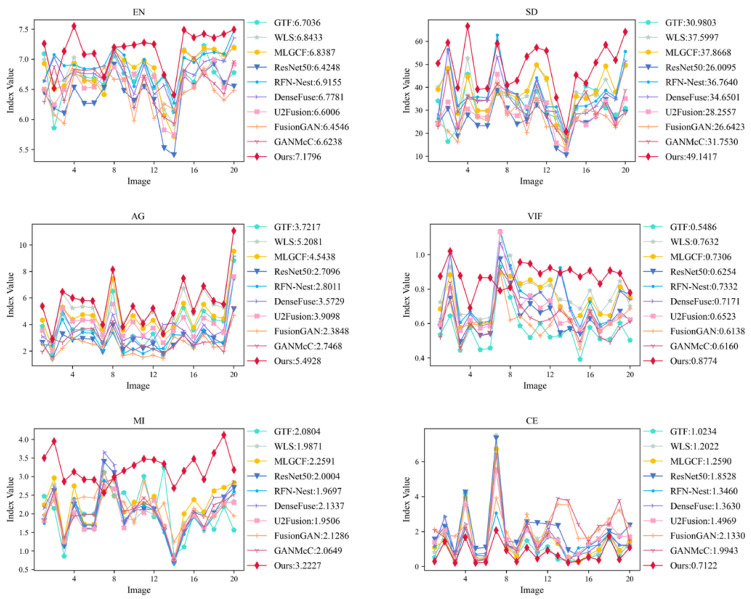
Twenty groups of infrared images and visible image different objective evaluation metric statistical broken line graph on the TNO dataset.

**Figure 11 entropy-24-01633-f011:**
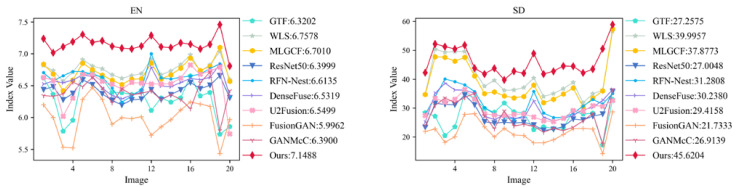
Twenty groups of infrared images and visible image different objective evaluation metric statistical broken line graph on the MSRS dataset.

**Table 1 entropy-24-01633-t001:** Validation of large-scale construction method.

	SD	MI	AG	CE
Weight Map	56.8479	14.5510	3.6330	0.4770
MWAO	57.2314	14.5519	4.0634	0.4578

**Table 2 entropy-24-01633-t002:** Ablation experiment setup and the average value of the evaluation metric of a fused image.

	ResNet50	FT++	FT	EN	SD	AG	VIF	MI	CE
(i)	-	√	-	7.1781	46.2157	4.7924	0.9593	3.9047	0.9390
(ii)	-	-	√	7.1761	48.6513	4.9052	0.9387	3.9015	0.7440
(iii)	-	-	-	6.8387	37.4755	4.5438	0.7360	2.2591	1.2590
(iv)	√	-	-	6.8216	35.9339	4.2283	0.7613	2.2544	1.5441
(v)	√	-	√	7.1637	48.2857	4.6031	0.9209	3.6287	0.7625
(vi)	√	√	-	7.1796	49.1417	5.4928	0.8774	3.2227	0.7122

## Data Availability

Not applicable.

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
