# Peer review of "Infrared and Visible Image Fusion with Significant Target Enhancement"

_entropy, 2022, doi:10.3390/e24111633_

Round 1

Reviewer 1 Report

The presentation of this paper is generally good and easy for readers to follow. I have some minor comments which the authors should address before the acceptance of this paper.

First, I find that some important references are missing in this paper. For example: Jia et al. [1] discuss how to perform image fusion for infrared and visible images. This study is directly related to this paper. Moreover, Li et al. [2] and Chen et al. [3] adopt the edge-preserving and deep-learning techniques to perform image fusion, respectively, which are also related to this paper. Therefore, the authors should discuss the differences between their methods and these methods [1-3]. In addition, there are also recent references in the image processing community, including [4-5], which are related to this paper. The authors should also discuss and cite them.

[1] Weibin Jia , Zhihuan Song, and Zhengguo Li "Multi-scale Fusion of Stretched Infrared and Visible Images" Sensors 2022

[2] Hui Li, Tsz Nam Chan, Xianbiao Qi, and Wuyuan Xie. "Detail-Preserving Multi-Exposure Fusion with Edge-Preserving Structural Patch Decomposition" TCSVT 2021

[3] Jie Chen, Zaifeng Yang, Tsz Nam Chan, Hui Li, Junhui Hou, and Lap-Pui Chau. "Attention-Guided Progressive Neural Texture Fusion for High Dynamic Range Image Restoration" TIP 2022

[4] Haoran Bai, Jinshan Pan, Xinguang Xiang, Jinhui Tang: "Self-Guided Image Dehazing Using Progressive Feature Fusion" TIP 2022

[5] Huafeng Li, Yueliang Cen, Yu Liu, Xun Chen, Zhengtao Yu: "Different Input Resolutions and Arbitrary Output Resolution: A Meta Learning-Based Deep Framework for Infrared and Visible Image Fusion" TIP 2021

Second, there are some English errors in this paper. The authors should proofread this paper carefully. Some examples include:

"The He et al. [18] proposed a" --> "He et al. [18] proposed a"

"Where $i \in{1,2,3}$ denotes" It should not use the capital letter for the word "where".

"as shown in Sections 4.1 and 4.2 respectively." --> there is a comma before the word "respectively".

"Subjective and objective evaluation of our method was" --> "Subjective and objective evaluations of our method were"

"The datasets are derived from TNO [24] and MSRS [25], respectively, and the selected images are aligned." You should not use the word "respectively" here.

Reviewer 2 Report

This manuscript presents a technically very interesting fusion method of infrared and visible images. The approach is very well substantiated by a very strong mathematical background. Simultaneously, experimental results, comparisons with other methods and objective and subjective evaluation are provided.  

The organization of the manuscript is given in an optimal way that is: reasoning, mathematical proof, performance assessment, comparisons, and conclusions. 

The literature review is adequate and the references up to date. 

I expect from the authors to add a paragraph at the end of section 1 for describing the rest sections and the organization of the paper. Furthermore, take care of the fonts of the heading like 4.3 and 5. 

Reviewer 3 Report

This paper proposes an infrared and visible image fusion method based on significant target enhancement. Experiments on two datasets validate the effectiveness of the proposed model. Detailed comments can be found as follows:

1. The motivations of the proposed method should be clearer. For example, the use of specific MLGCF, etc.

2. The third contribution is related to experiments, which is suggested to merge into other contributions.

3. The equations in Figure 2 seem to have no explanations. Please add more statements.

4. Some other infrared image processing tasks are recommended to be reviewed, including Infrared pedestrian detection with converted temperature map, Infrared image segmentation based on Otsu and genetic algorithm, etc.

5. In the experiments, the ablation study of FT and FT++ should be compared to further verify the proposed method.

Round 2

Reviewer 3 Report

The authors have addressed all my comments.